DISCOVERY REPORT

# *Pseudomonas aeruginosa* cleaves the decoding center of *Caenorhabditis elegans* ribosomes

**Alejandro Vasquez-Rifo** [1] *, **Emiliano P. Ricci** [2], **Victor Ambros** [1] *

**1** Program in Molecular Medicine, University of Massachusetts Medical School, Worcester, Massachusetts, United States of America, **2** Laboratoire de Biologie et Modélisation de la Cellule, Université de Lyon, École normale supérieure de Lyon, Université Claude Bernard Lyon 1, CNRS UMR 5239, INSERM U1210 Lyon, France

* alejandro.vasquezrifo@umassmed.edu (AV-R); victor.ambros@umassmed.edu (VA)

**Data Availability Statement:** All relevant data are within the paper and its supporting information

## Abstract

Pathogens such as *Pseudomonas aeruginosa* advantageously modify animal host physiology, for example, by inhibiting host protein synthesis. Translational inhibition of insects and mammalian hosts by *P. aeruginosa* utilizes the well-known exotoxin A effector. However, for the infection of *Caenorhabditis elegans* by *P. aeruginosa*, the precise pathways and mechanism(s) of translational inhibition are not well understood. We found that upon exposure to *P. aeruginosa* PA14, *C. elegans* undergoes a rapid loss of intact ribosomes accompanied by the accumulation of ribosomes cleaved at helix 69 (H69) of the 26S ribosomal RNA (rRNA), a key part of ribosome decoding center. H69 cleavage is elicited by certain virulent *P. aeruginosa* isolates in a quorum sensing (QS)–dependent manner and independently of exotoxin A–mediated translational repression. H69 cleavage is antagonized by the 3 major host defense pathways defined by the *pmk-1*, *fshr-1*, and *zip-2* genes. The level of H69 cleavage increases with the bacterial exposure time, and it is predominantly localized in the worm's intestinal tissue. Genetic and genomic analysis suggests that H69 cleavage leads to the activation of the worm's *zip-2*-mediated defense response pathway, consistent with translational inhibition. Taken together, our observations suggest that *P. aeruginosa* deploys a virulence mechanism to induce ribosome degradation and H69 cleavage of host ribosomes. In this manner, *P. aeruginosa* would impair host translation and block antibacterial responses.

## Introduction

Pathogens deploy diverse virulence mechanisms to exploit host resources and to counteract or evade host defenses. The host translational machinery, including the ribosome and associated factors, is frequently targeted in such host–pathogen interactions [1].

Pathogenic bacteria benefit from repression of host protein synthesis, through accelerated cellular damage and blockage of host immune responses. Instances of host translation targeting by pathogenic bacteria include those elicited by *Corynebacterium diphtheriae*, *Vibrio cholerae*, and *Pseudomonas aeruginosa*, which encode bacterial toxins that ADP-ribosylate and inhibit the eukaryotic translation elongation factor 2 [2,3]. Similarly, *Legionella pneumophila*

files (S1 Data and S1–S4 Tables). Sequencing data
files are available on Github at https://github.com/
avasrifo/Paecleavage/.

**Funding:** This research was supported by funding
from NIH grants R01GM088365, R01GM034028
and R35GM131741 (V.A.) and the Pew Charitable
Trusts (A.V-R.). Some C. elegans strains were
provided by the CGC, which is funded by NIH Office
of Research Infrastructure Programs (P40
OD010440). The funders had no role in study
design, data collection and analysis, decision to
publish, or preparation of the manuscript.

**Competing interests:** The authors have declared
that no competing interests exist.

**Abbreviations:** ER, endoplasmic reticulum; GPCR,
G protein-coupled receptor; H69, helix 69; H79,
helix 79; IP, immunoprecipitation; LB, Lysogeny
Broth; lf, loss of function; nt, nucleotides; PAGE,
polyacrylamide gel electrophoresis; QS, quorum
sensing; RNAi, RNA-mediated interference; RPM,
reads per million; rRNA, ribosomal RNA; SK, slow
killing; ts, temperature sensitive; UPR$^{ER}$, unfolded
protein response of the endoplasmic reticulum;
UPR$^{mt}$, mitochondrial unfolded protein response;
YA, young adults.

toxins inhibit protein synthesis by glucosylation of elongation factor 1A [4]. A different mechanism of translational repression is employed by *Shigella dysenteriae* and *Escherichia coli* strains O157:H7 and O104:H4. These bacteria encode toxins that depurinate a specific adenine in the 28S ribosomal RNA (rRNA) [5].

*P. aeruginosa* is a free-living proteobacteria that possesses the ability to infect a wide range of animal hosts, including humans [6–8]. A major *P. aeruginosa* effector that mediates translation inhibition for many hosts is the exotoxin A protein (encoded by the *toxA* gene) [6,9,10]. ToxA can inhibit translation in *Caenorhabditis elegans* [11]. However, ToxA is likely not the only translational inhibitory activity in *P. aeruginosa* for *C. elegans*, since mutation of ToxA does not appreciably abrogate a host response to translation inhibition and does not alter virulence towards wild-type *C. elegans* [11]. In this study, we addressed whether *P. aeruginosa* has multiple strategies for host translation inhibition and investigated the effects of *P. aeruginosa* on the ribosomes of the nematode *C. elegans*.

The interaction between *C. elegans* and the virulent *P. aeruginosa* strain PA14 is a well-studied model host–pathogen interaction. Under the so-called slow killing (SK) coculture conditions, PA14 bacteria colonize the intestine of adult worms and kill the worms in the course of approximately 3 days [12]. Under these conditions, the bacteria engage virulence-promoting gene expression programs through the quorum sensing (QS) regulatory pathways [13]. The bacterial QS pathway is integral to the pathogenicity of *P. aeruginosa* against a broad range of animal hosts, including vertebrates and nematodes such as *C. elegans* [12,14].

In response to encountering *P. aeruginosa*, the nematode activates multiple behavioral, stress response, and innate immune pathways to counter PA14-induced damage [15–19]. These include the *pmk-1*/P38 MAP kinase pathway [15], the *fshr-1*/G protein-coupled receptor (GPCR) pathway [17], and the *zip-2* pathway, which is elicited in response to repression of translation [11,19,20].

In this paper, we establish that upon interaction with virulent *P. aeruginosa* strains, *C. elegans* experiences a drastic loss of rRNA content and a rapid accumulation of ribosomes with the 26S rRNA cleaved at the decoding center in the highly conserved helix 69 (H69) hairpin. Our results indicate that this previously unknown 26S rRNA cleavage at H69 is not a generic response to stress or cell death but rather reflects a virulence mechanism specific to certain *P. aeruginosa* isolates. H69 cleavage depends on gene regulators of the bacterial virulence program, including QS, and is independent of *toxA*-mediated translational repression. H69 cleavage is antagonized by 3 PA14-activated *C. elegans* antipathogen response pathways, including the *zip-2* pathway, which is known to be activated in response to translational inhibition. Consistent with translational inhibition by H69 cleavage in *C. elegans*, genetic perturbations that abrogate H69 cleavage by PA14 (e.g., *C. elegans dyn-1* or *P. aeruginosa gacA*) also prevent *zip-2* pathway activation. Furthermore, the ability of individual *P. aeruginosa* wild strains to elicit H69 cleavage precisely correlates with each strain's activation of the worm's *zip-2* pathway. Based on our findings, we propose that ribosome cleavage at H69 reflects a novel virulence mechanism that *P. aeruginosa* deploys to inhibit translation in animal hosts.

## Results

### *C. elegans* exposed to *P. aeruginosa* strain PA14 accumulate ribosomes cleaved at H69

In order to study the effect of exposure to *P. aeruginosa* strain PA14 (referred to as PA14) on the worm ribosomes, we analyzed by capillary electrophoresis the total RNA profile of adult hermaphrodites fed either with *E. coli* or with *P. aeruginosa* strain PA14. Total RNA extracted from adults after 24 h on PA14-seeded SK plates exhibited 2 prominent RNA bands of

approximately 1,100 nucleotides (nt) and 2,300 nt (Fig 1A) that were not observed in the control condition of worms feed with *E. coli* HB101. To identify the origins of these novel species, total worm RNA was fractionated by gel electrophoresis, and approximately 1,100 nt and approximately 2,300 nt bands were excised and ligated to linkers at their 3′ and 5′ ends. cDNA sequencing revealed that the 2 bands correspond to 26S rRNA. The 5′ terminus of the approximately 2,300 band coincided with the 5′ terminus of full-length 26S rRNA, and its 3′ terminus localized to nt position 2,360 of the 26S rRNA. The 5′ terminus of the approximately 1,100 band localized to 26S rRNA nt position 2,362, and its 3′ terminus coincided with the 3′ terminus of the full-length 26S rRNA (Fig 1B). These results indicate that the 2 rRNA fragments observed in total RNA from PA14-exposed worms correspond to fragments of the full-length 26S rRNA cleaved at nucleotide positions 2,360 and 2,362 (Fig 1B). The fragment cut sites are localized to the loop of H69, a highly conserved structural element that constitutes part of the ribosome decoding center and that by interaction with helix 44 forms part of the inter-subunit bridge B2a [21].

Quantitative capillary electrophoresis was used to monitor the accumulation of the two 26S fragments in samples of total RNA from worms exposed to PA14. The level of accumulation of the two 26S fragments (i.e., H69-cleaved rRNA) was measured relative to the total 26S rRNA pool (fragments plus full length). By 24 h of PA14 exposure, an average 25% of the worm 26S rRNA consisted of cleaved fragments (Fig 1C, S1 Data). In contrast, no H69-cleaved rRNA was apparent in worms under normal growth conditions (i.e., on *E. coli* HB101).

rRNA fragmentation (albeit not necessarily at H69) has been reported to occur in cells exposed to stress conditions [22–25]. Thus, we evaluated whether accumulation of H69-cleaved rRNA fragments could be a common consequence of exposure of worms to pathogens or stress conditions. Exposure of *C. elegans* to other lethal bacterial pathogens, including *Staphylococcus aureus*, *Salmonella enterica*, or *Enterococcus faecalis*, did not result in accumulation of H69-cleaved RNA (S1A Fig). Similarly, treatment of worms with an array of other stressors did not recapitulate the PA14-induced rRNA cleavage (S1A Fig).

Concomitantly with the accumulation of H69-cleaved RNA, worms cultured on *P. aeruginosa* PA14 also display a dramatic reduction in total RNA content per worm (S1B Fig, S1 Data). Although a similar global RNA loss was observed for *C. elegans* adults that were exposed to another pathogen, *E. faecalis*, or that were starved for 24 h, these 2 other conditions did not cause any evident H69 cleavage (S1B Fig). Thus, the accumulation of H69-cleaved 26S rRNA fragments in PA14-infected worms does not seem to occur as a consequence of processes associated with a generalized pathogen-induced or starvation-induced reduction in rRNA content.

To further confirm that H69 cleavage is specific to worms exposed to *P. aeruginosa*, we sought to test whether the H69 cleavage might occur in worms cultured on *E. coli* HB101 at levels below the sensitivity of capillary or gel electrophoresis. Accordingly, we employed "degradome" sequencing of total worm RNA to catalog the positions of the 5′ ends of rRNAs (Fig 1D). For *C. elegans* adults cultured on *E. coli* HB101 for 24 h, most 26S rRNA termini mapped to the annotated 5′ end of the RNA, and the frequency of termini at H69 was no more abundant than the background level of termini across the length of 26S rRNA (Fig 1D). In adults treated with PA14 for 24 h, 26S rRNA 5′ termini at H69 occurred well above background (13% of the total reads mapping to 26S), consistent with a pathogen-induced specific cleavage of 26S rRNA at H69. We also noted an accumulation of termini corresponding to the position of helix 79 (H79) in 26S rRNA from worms cultured on PA14 (Fig 1D), indicating at least 1 additional *P. aeruginosa*–induced cleavage in 26S RNA. Of note, 18S, 5S, and 5.8S rRNA molecules did not show any evident accumulation of internal termini upon exposure to PA14 (S3 Fig).

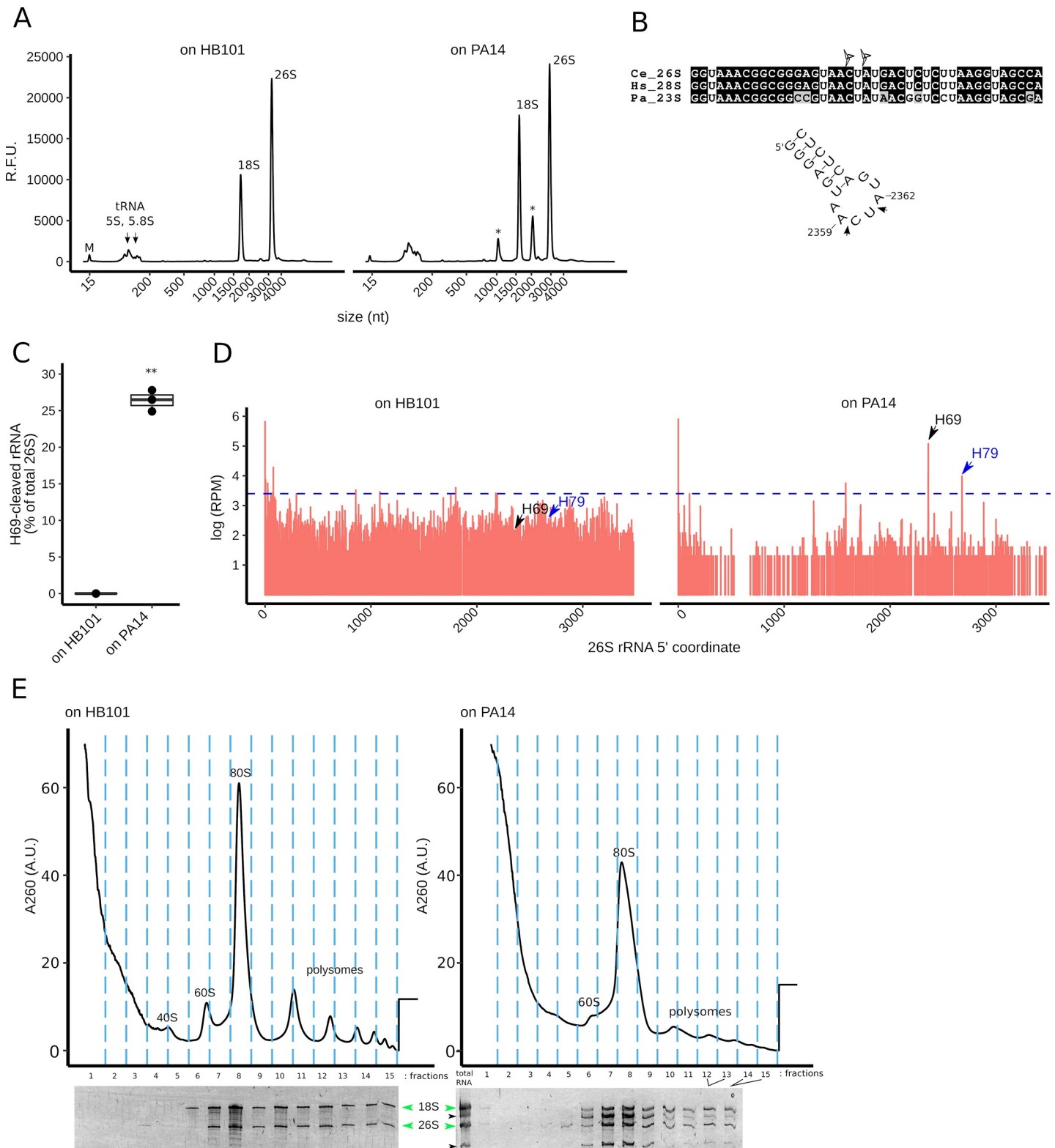

**Fig 1. Effects of *P. aeruginosa* PA14 on the rRNA of *C. elegans* worms. (A)** Total RNA profile of adult worms exposed for 24 h to *P. ae.* PA14 (right panel) or *Escherichia coli* HB101 (left panel). Asterisks indicate the 2 most abundant bands that appear upon PA14 exposure. "M" indicates a 15-nt marker used in the electrophoretic separation system. **(B)** Top: 26S rRNA alignment of *C. elegans* (*Ce*), *Homo sapiens* (*Hs*), and *P. aeruginosa* (*Pa*) sequences around the cut sites (triangles) that define the 2 bands indicated on (A). Bottom: secondary structure of the region comprising the cut sites (arrowheads). The hairpin structure corresponds to the 26S

rRNA H69. **(C)** H69 cleavage levels (as percentage of the total 26S rRNA) in adult worms exposed to PA14 or HB101 for 24 h. "**" represents a *p*-value < 0.01 for a Welch *t* test comparison between the plotted conditions. S1 Data contains source data for panel C. **(D)** Abundance of 5′ termini along the worm 26S rRNA (in log10 of RPM) as determined by degradome sequencing. Two conditions are shown: worms on *P. ae*. PA14 (right panel) or worms on control *E. coli* HB101 (left panel). The H69 5′ terminus is indicated by the black arrowhead. The dashed blue line marks the level of abundance below, which cuts are considered to be at background level (i.e., outliers in the cut site frequency distribution). Blue arrows indicate the cleavage site in H79. **(E)** Representative polysome profiling of worm ribosomes upon 24-h exposure to *E. coli* HB101 (left panel graph) or PA14 (right panel graph). The corresponding RNA of each gradient's fraction was separated on a denaturing PAGE gel. The bands corresponding to the H69 fragments are indicated by black arrowheads. A sample of total RNA from PA14-exposed worms is shown (right panel, "total RNA" lane). Intact 18S and 26S rRNAs are indicated by green arrows. H69, helix 69; H79, helix 79; PAGE, polyacrylamide gel electrophoresis; R.F.U. relative fluorescence units; A.U. arbitrary units; *P. ae*. PA14, *P. aeruginosa* PA14; RPM, reads per million; rRNA, ribosomal RNA.

To determine whether cleavage of 26S rRNA at H69 occurs within intact ribosomes and/or 60S subunits, we fractionated extracts of worms on sucrose gradients to separate 40S and 60S ribosomal subunits, 80S monosomes, and polyribosomes and analyzed the RNA from each gradient fraction (Fig 1E). For adults cultured on *E. coli* for 24 h, the expected 18S and 26S RNAs sedimented with the free small and large ribosomal subunits, as well as with 80S monosomes and larger polyribosomes. For worm adults cultured on PA14 for 24 h, a reduced profile of polyribosomes relative to monosomes was evident, indicative of PA14-induced translational inhibition. H69-cleaved fragments of 26S RNAs appeared predominantly in the fractions containing 80S monosomes, to a lesser extent in the polyribosomal fraction, and were absent from the ribosomal-free light gradient fractions. These results support the conclusion that H69 cleavage of 26S RNA occurs in intact large ribosomal subunits contained within 80S ribosomes.

To test for the presence of an activity associated with PA14-infected worms that can elicit H69 cleavage to intact *C. elegans* ribosomes, we sought to recapitulate the generation of the approximately 2,300 and approximately 1,100 26S rRNA fragments using an in vitro assay. PA14-exposed worms were used to prepare an rRNA-free S100 extract. The S100 extract was incubated with ribosomes isolated from unexposed animals, and rRNA was extracted and resolved by electrophoresis. The S100 from PA14-exposed worms caused the appearance of 2 bands (approximately 1,100 and approximately 2,300 nt) on the substrate ribosomes (S2A Fig). The bands were of the same size as those observed in vivo. cDNA prepared from these bands was sequenced and confirmed to correspond to fragments of 26S rRNA with termini identical to those observed in vivo (S2B Fig). No similar activity was detected in S100 from *C. elegans* exposed to *E. coli* (S2A Fig) These results support the hypothesis that PA14-infected *C. elegans* contain an activity that can cleave the worm's 26S rRNA within the intact ribosomal large subunit.

## H69 cleavage requires bacterial virulence pathways and is independent of *toxA*

The expression of *P. aeruginosa* genes required for virulence against an animal host such as *C. elegans* is controlled by major virulence regulators, including QS, and the 2-component system *gacA*/*gacS* pathway. We observed that mutations in the QS genes *lasR* or *rhlR*, or in *gacA*, or the transcriptional regulatory protein PA14_27700, disabled the capacity of PA14 to induce H69 cleavage (Fig 2A, S1 Data). We conclude that the bacterial capacity to induce rRNA cleavage is controlled by the gene regulatory network that governs the expression of bacterial virulence, suggesting that the cleavage reflects a bacterial strategy aimed at compromising the host's defenses against infection.

*P. aeruginosa* strains with a broad range of virulence against *C. elegans* have been described [26,27]. To test for a correlation between virulence and H69 cleavage, we screened a panel of 52 *P. aeruginosa* isolates that exhibit a wide range of virulence. We found that only certain

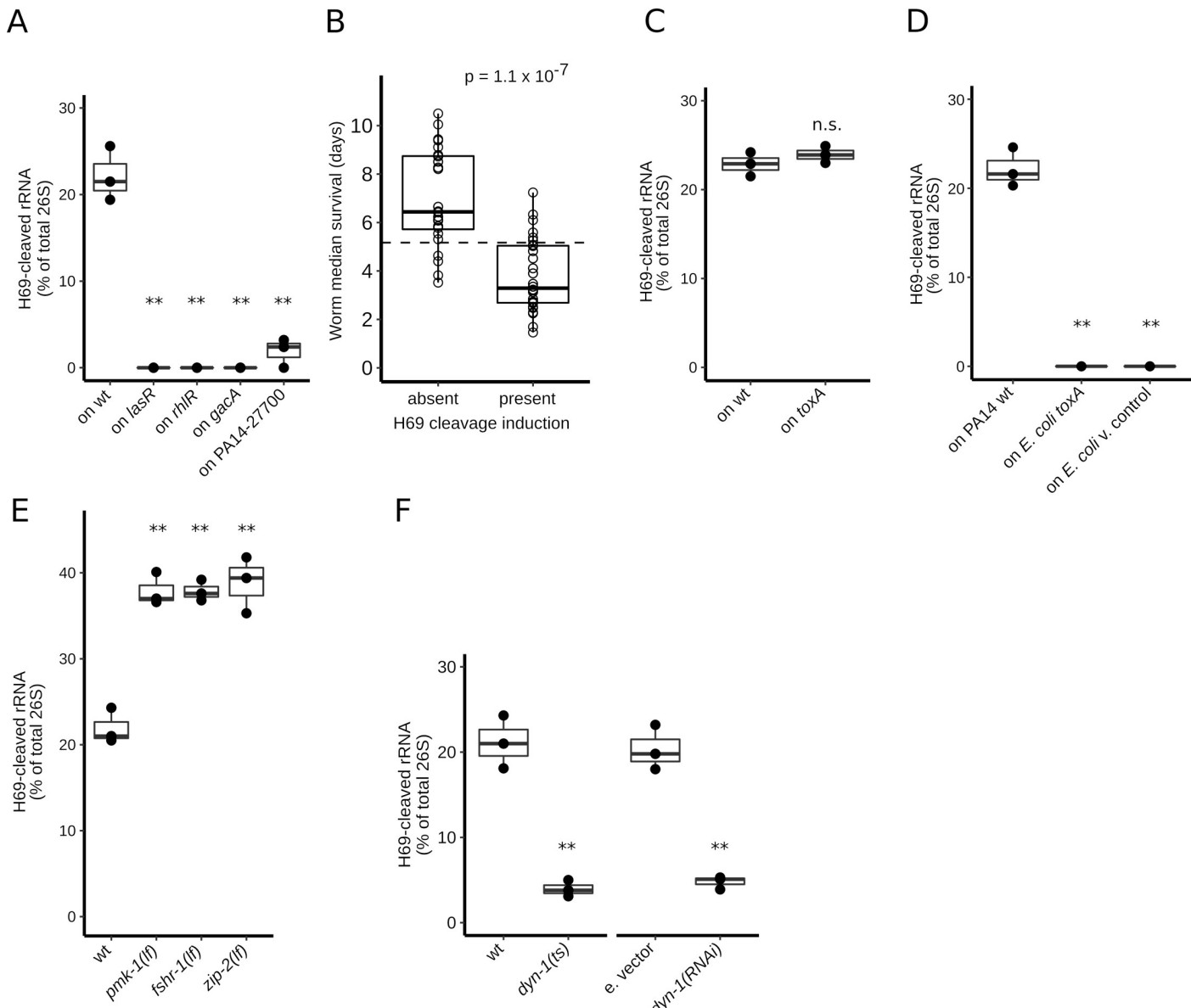

**Fig 2. Relationship between H69 cleavage and virulence, virulence genes, and host genes. (A)** H69 cleavage levels for *Caenorhabditis elegans* worms exposed 24 h to 4 PA14 mutants (*lasR*, *rhlR*, *gacA*, and PA14_27700). All mutants are required for virulence towards *C. elegans* and 3 of them regulate bacterial QS (*lasR*, *rhlR*, and *gacA*). **(B)** Relationship between the capability to induce H69 cleavage and *Pseudomonas aeruginosa* virulence. The virulence of *P. aeruginosa* wild isolates is represented by the median survival time (in days) of wt worms exposed to the isolate, as determined previously [27]. H69 cleavage–inducing capacity (present or absent) was determined by analysis of total RNA from worms exposed to each isolate for 24 h, using capillary electrophoresis. The mean virulence of the 2 isolate sets is significantly different (Welch *t* test, *p*-value = $1.1 \times 10^{-7}$). S3 Table contains source data for panel B. **(C)** H69 cleavage levels for worms exposed to a bacterial mutant of exotoxin A (*toxA*) or wt PA14 for 24 h. The cleavage levels are n.s. (Welch *t* test, *p*-value > 0.05). **(D)** H69 cleavage levels for worms exposed to *E. coli* expressing the *toxA* gene or v. control for 24 h. H69 cleavage for PA14-exposed worms is included as control. **(E)** H69 cleavage levels of *C. elegans lf* mutants in 3 host response pathways (*pmk-1*, *fshr-1*, and *zip-2*). The worms were exposed to PA14 for 24 h. **(F)** H69 cleavage levels, following exposure to PA14 for 24 h, for *C. elegans* worms with reduced dynamin (*dyn-1*) gene function. Two experimental conditions are shown: (1) a *dyn-1 ts* (restrictive at 25°C) is compared to wt worms; and (2) *dyn-1* RNAi knockdown in the intestine (using strain VP303) is compared with e. vector RNAi condition. For all graphs, "*" indicates *p*-value < 0.05, and "**" represents *p*-value < 0.01 for Welch *t* test comparisons between the plotted conditions. S1 Data contains source data for panels A and C–F. e. vector, empty vector; H69, helix 69; *lf*, loss of function; n.s., not significantly different; QS, quorum sensing; RNAi, RNA-mediated interference; *ts*, temperature sensitive; v. control, vector control; wt, wild-type.

strains in the panel are able to induce H69 cleavage (S1C Fig). Moreover, we observed a strong correlation between strain virulence, as measured by the mean life span of adults exposed to each strain and the occurrence of H69 cleavage (Welch $t$ test, $p$-value = $1.1 \times 10^{-7}$, Fig 2B, S3 Table). This result further supports the idea that H69 cleavage is a component of *P. aeruginosa* pathogenicity. Previous studies showed that *P. aeruginosa* ToxA can inhibit translation in *C. elegans* when expressed in the worms' *E. coli* food, but for *P. aeruginosa* PA14, ToxA is likely not the only translational inhibitory activity [11]. We observed that a *toxA* mutant of PA14 elicited normal levels of H69 rRNA cleavage (Fig 2C, S1 Data). Moreover, exposure of *C. elegans* to *E. coli* bacteria expressing ToxA did not cause H69 cleavage (Fig 2D, S1 Data). These results indicate that *toxA* is not necessary for the occurrence of H69 cleavage. Thus, H69 cleavage of *C. elegans* rRNA by *P. aeruginosa* entails a *toxA*-independent pathway.

If not ToxA, what other PA14 virulence factor(s) could be responsible for executing H69 cleavage? Our results described above (Fig 2A) show that H69 cleavage depends on the activity of the major regulators PA14 virulence, including QS (*lasR* and *rhlR*) and 2-component signaling (*gacA/gacS*). Therefore, we tested whether H69 cleavage could be attributed to well-known virulence-related factors downstream of *lasR/rhlR* and *gacA/gacS*. For all the PA14 mutants we tested (*exoU*; genes of the pyoverdin, rhamnolipid, phenazine, and pyochelin biosynthetic pathways; types I, II, III, V, and VI secretion systems; S4 Fig), H69 cleavage occurred normally. Therefore, we conclude that H69 cleavage depends on unidentified virulence factors downstream of the QS and 2 component regulators.

## H69 cleavage is antagonized by host response pathways and requires dynamin function

*C. elegans* activates a set of defense pathways upon exposure to PA14, enabling the worm to mount stress responses and immune defense. These defense pathways include the *pmk-1*/P38 MAP kinase pathway [15], the *fshr-1*/GPCR pathway [17,28], and the *zip-2* pathway, which is elicited in response to repression of translation [11,19,20]. *C. elegans* mutants with loss-of-function (*lf*) mutations in *pmk-1*, *fshr-1*, or *zip-2* have increased sensitivity to PA14 infection compared to the wild type [15,17,19,28]. Likewise, we observed an increase in H69 cleavage upon exposure of *pmk-1*, *fshr-1*, or *zip-2* mutants to PA14 (Fig 2E, S1 Data). This finding supports the conclusion that rRNA H69 cleavage in worms exposed to PA14 results from a bacterial virulence mechanism that is opposed by the action of the host defense pathways involving *pmk-1*, *fshr-1*, and *zip-2*.

Activation of the *zip-2* pathway by PA14 has been shown to require components of the *C. elegans* endocytic machinery, including the dynamin protein DYN-1 (encoded by the *dyn-1* gene) [20]. The requirement of *dyn-1* for *zip-2* activation suggests that bacterial factors that inhibit translation are internalized into worm cells by endocytosis [20]. Thus, we tested whether H69 cleavage requires *dyn-1* activity. Interestingly, the *dyn-1* temperature-sensitive *(ts)* mutant shows dramatically reduced levels of ribosome cleavage upon PA14 exposure compared to the wild type (Fig 2F, S1 Data). Similarly, intestine-specific *dyn-1* knockdown was sufficient to abrogate cleavage (Fig 2F). These results suggest that dynamin-mediated endocytosis is required for the uptake of 1 or more *P. aeruginosa* factors that elicit rRNA cleavage.

## Accumulation of H69-cleaved ribosomes tracks with *P. aeruginosa* exposure and is predominately localized in the intestine

The levels of H69-cleaved RNA in adults exposed to PA14 increase linearly with time of exposure (Fig 3A, S1 Data). Cleavage fragments can be detected by capillary electrophoresis of total worm RNA after 8 h of exposure, and the amount of cleavage fragments increases to

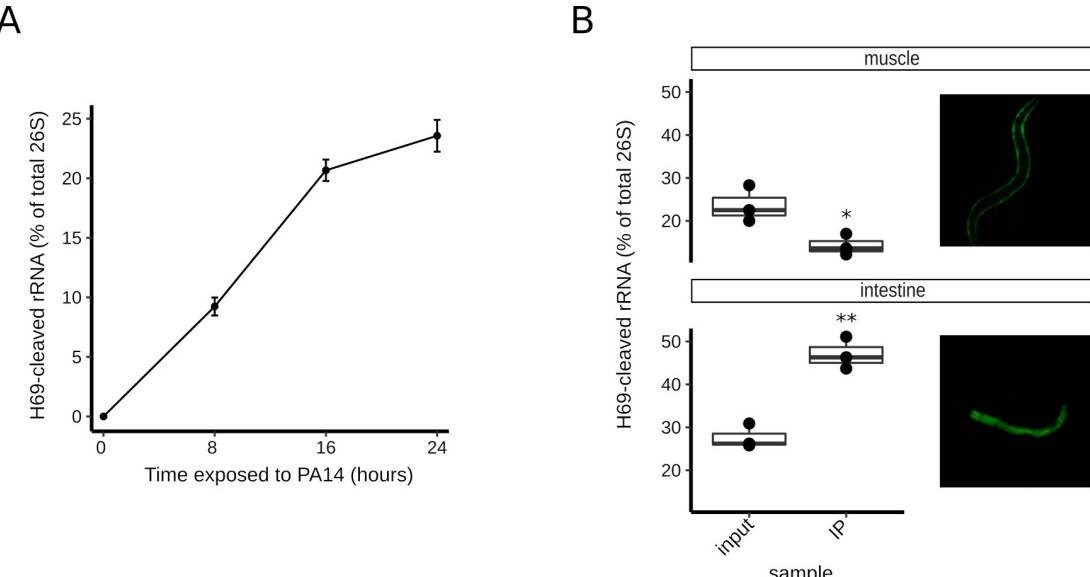

**Fig 3. Dynamics of H69 cleavage. (A)** Graph of H69 cleavage levels in relationship to adult worm exposure time to PA14 (in hours). The average of tree measurements per timepoint and SD (error bars) is plotted. **(B)** H69 cleavage levels in 2 major worm tissues. Worms with GFP-tagged ribosomes in the muscle (top panel, JAC127 strain) and intestine (bottom panel, JAC382 strain) were exposed to PA14 for 24 h, followed by tagged ribosome IP and RNA analysis. The H69 cleavage levels for whole worms (labeled "input") and the corresponding "IP" ribosomes are plotted. For all graphs, "*" indicates *p*-value < 0.05, and "**" represents *p*-value < 0.01 for Welch *t* test comparisons between the plotted conditions. S1 Data contains source data for all panels. GFP, green fluorescent protein; H69, helix 69; IP, immunoprecipitation; SD, standard deviation.

approximately 25% of the whole worm 26S rRNA by 24 h. Adults completely succumb to PA14 infection and die in the course of approximately 3 days (with a median survival of 2.1 days) [26,27]. The 24-h exposure was selected as a standard timepoint for rRNA analyses, as essentially all worms are still alive at that timepoint. The observation that H69 cleavage is detected well before the first animals in the population die suggests that cleavage is not a consequence of a process coupled to death of the animal. Similarly, we found that worms fed *E. coli*–expressing Cry6A, a toxin that induces necrosis [29], did not exhibit H69 cleavage (S1A Fig). Also, cellular apoptosis appears not to be required for H69 cleavage, as a *ced-3(lf)* mutant, which is apoptosis defective [30], displayed normal levels of H69 cleavage upon PA14 exposure (S5A Fig, S1 Data).

In contrast to PA14-fed C. *elegans* adults, larvae are less susceptible to PA14-mediated killing; indeed, larvae are fed with PA14 as their sole food source readily complete the entirety of development, albeit at a somewhat reduced pace. Interestingly, H69 cleavage is observed in larvae exposed to PA14 (S5B Fig, S1 Data), although at a lower level than in PA14-fed adults. Apparently, the H69 cleavage experienced by PA14-fed larvae is sufficient to perhaps contribute to slowing the rate of larval developmental but is otherwise fully compatible with larval viability and development to the adult. This result also shows that H69 cleavage elicited by PA14 can occur in the absence of overt organismal death.

H69 cleavage could potentially reflect a systemic response to PA14 in most or all tissues of the animal, or alternatively, a cell-intrinsic process associated with tissues that most directly interact with the bacterium, which is primarily the intestine. To investigate the occurrence of H69 cleavage in distinct worm tissues, transgenic worms that express green fluorescent protein (GFP)-tagged ribosomes specifically in the intestine or body muscles [31] were exposed to PA14, followed by ribosome immunoprecipitation (IP) and analysis of the

immunoprecipitated rRNA. H69 cleavage was found to occur at lower levels in muscle compared to whole worms (Fig 3B, S1 Data). In contrast, cleavage was highly enriched in the intestine, where its prevalence was approximately twice than that measured in whole worms. These results suggest that H69 cleavage occurs predominantly in the worm intestinal cells that are in direct contact with PA14 bacteria.

## H69 cleavage induces a *zip-2*-mediated host response to translation repression

Mutations in H69 have been shown to be lethal in vivo in *E. coli* [32–35] and impair in vitro translation [32]. Consequently, we expected that H69-cleaved *C. elegans* ribosomes would similarly be defective for translation. Translational inhibition of *C. elegans*, for example, by exposure to the drugs cycloheximide or hygromycin, leads to the activation of the *zip-2* translational inhibition response pathway, including the preferential translation of *zip-2* mRNA, leading to accumulation of ZIP-2, a bZIP transcription factor that in combination with CEBP-2 [36], guides transcription of target genes [19,36]. The *zip-2* pathway mediates expression of target genes that are regulated mostly independently of other stress pathways and encompass detoxification genes such as the p-glycoprotein-related *pgp-5* and *pgp-7* [37] (presumably to help neutralize translational inhibitory xenobiotics) and other less well-understood genes such as *irg-1* and *irg-2* [19].

Expression of an *irg-1*::*GFP* transgene in *C. elegans* can be used to gauge the activation of the *zip-2* pathway, and thereby infer a likely response to protein synthesis inhibition [11,19]. PA14 exposure has been shown to elicit translation inhibition (Fig 1E) [11,19] and *zip-2* pathway activation in *C. elegans* [11,19]. We posit that the H69 cleavage, which occurs at high levels in intestinal cells, would lead to significant translation impairment and *zip-2* pathway activation. Under this scenario, we expect that perturbations that modulate H69 cleavage should be paralleled by changes in *irg-1*::*GFP* expression.

Consistent with previous findings that mutation of PA14 *toxA* does not appreciably abrogate *zip-2* pathway activation in *C. elegans* [11], we observed that the PA14 *toxA* mutant elicited normal induction of *irg-1*::*GFP* expression (Fig 4A) and normal levels of H69 cleavage of 26S rRNA (Fig 2C). In contrast, impairment of PA14 virulence with the *gacA* mutant abrogated both *irg-1*::*GFP* expression (Fig 4B) and H69 cleavage (Fig 2A). A third perturbation, intestinal knockdown of *dyn-1* by RNA-mediated interference (RNAi), abrogated H69 cleavage (Fig 2F) and hampered *irg-1*::*GFP* expression compared to the wild type (Fig 4C). These results support the model that H69 cleavage induces translation repression and hence *zip-2* pathway activation.

To further assess the contribution of H69 cleavage to translational repression and virulence of *P. aeruginosa* for *C. elegans*, we tested the association between H69 cleavage and activation of *zip-2* pathway across a panel of 26 *P. aeruginosa* isolates of known virulence. Adult worms carrying the *irg-1*::*GFP* transgene were exposed to each strain, and GFP fluorescence was quantified after 24 h (Fig 4D and 4E, S3 Table). Strikingly, all strains that elicited H69 26S rRNA cleavage (14 isolates) also induced the *irg-1* transgene. Conversely, all strains unable to evoke ribosome cleavage (12 isolates) had negligible *irg-1*::*GFP* transgene expression. These results indicate a strong statistical association ($\chi^2$ test, *p*-value = $2.5 \times 10^{-6}$) between rRNA H69 cleavage and *zip-2* pathway activation (i.e., *irg-1*::*GFP* expression).

To determine the specificity of the correlation between cleavage and *zip-2* pathway activation, 3 other host pathways implicated in the response to PA14 [15,38–40] were tested for association with H69 cleavage. The endoplasmic reticulum (ER) unfolded protein response (UPR^ER) [39,40], the mitochondrial unfolded protein response (UPR^mt) [38], and the *pmk-1/*

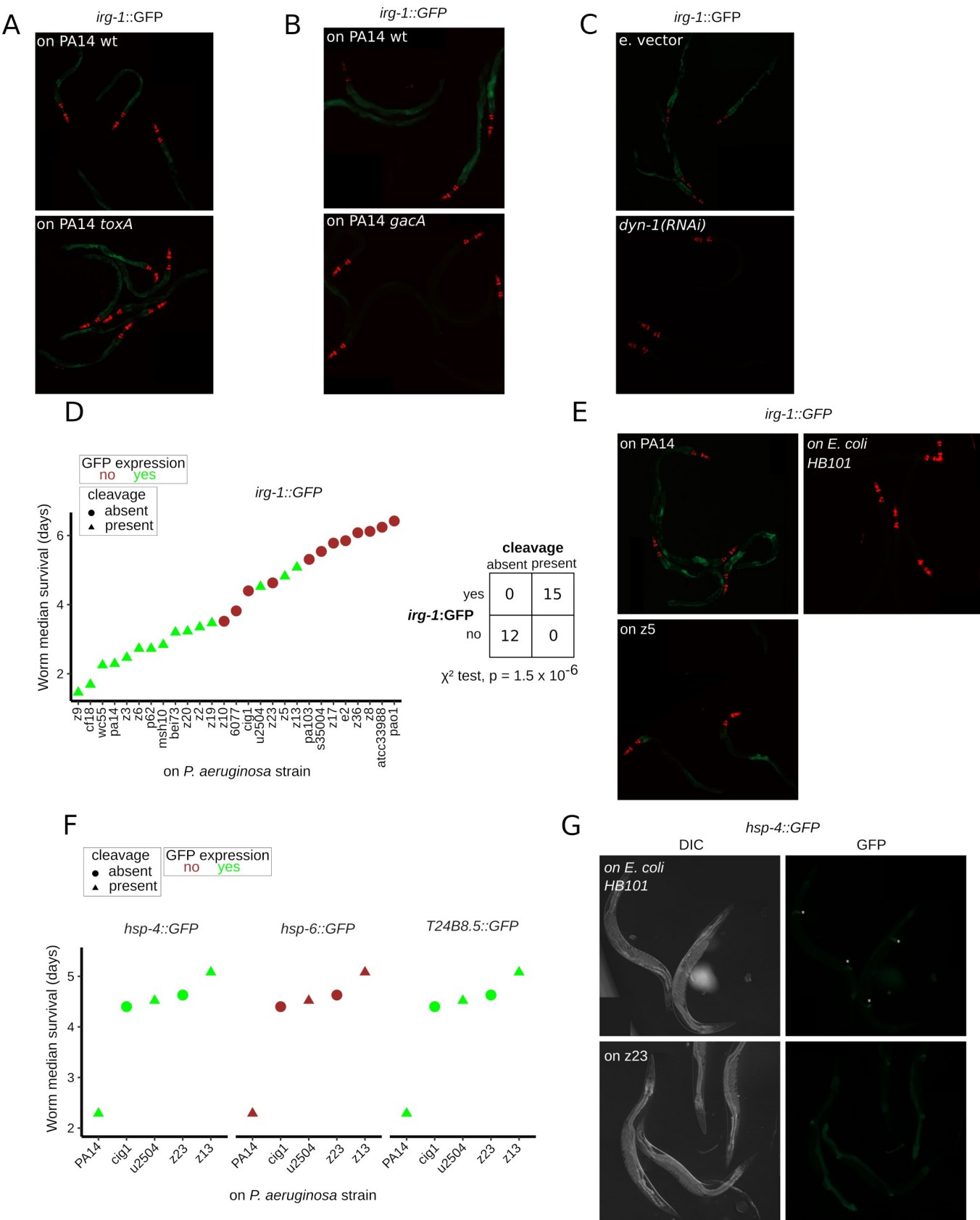

**Fig 4. Relationship between H69 cleavage and the *zip-2* pathway. (A and B)** Fluorescent microscopy images of *irg-1*::*GFP* worms (strain AU133) exposed to (1) PA14 *toxA* and PA14 wt for 12 h (A); and (2) PA14 *gacA* and PA14 wt for 12 h (B). **(C)** Fluorescent microscopy images of *irg-1*::GFP worms (strain VT4003) exposed to PA14 for 12 h, after intestinal specific *dyn-1* knockdown (*dyn-1* (RNAi)) or treatment with e. vector RNAi. **(D)** Top: relationship between the bacterial capacity to induce H69 cleavage (absent or present, in circles and triangles, respectively) and the capacity to induce expression of *irg-1*::GFP (absent or present, indicated by red and green colors, respectively) among *Pseudomonas aeruginosa* wild isolates. *irg-1*::*gfp* worms were exposed to each bacterial isolate, GFP expression was evaluated at 12 and 24 h of exposure with identical results, and worm RNA analysis was conducted at 24 h of exposure. The isolates are ranked by virulence (i.e., median worm survival in days). Bottom: contingency table for the H69 cleavage and *irg-1*::GFP-inducing capacities of the *P. aeruginosa* isolates. The 2 outcomes are significantly associated ($p$-value = $1.5 \times 10^{-6}$, $\chi^2$ test). **(E)** Fluorescent microscopy images of *irg-1*::GFP worms exposed to *E. coli* HB101 (nonpathogenic control) or *P. aeruginosa* wild isolates PA14 or z5. Adult worms were exposed to the respective bacteria for 12 h. **(F)** Relationship between the bacterial capacity to induce H69 cleavage and the expression of *hsp-4*::GFP, *hsp-6*::GFP, and *T24B8.5*::GFP reporters. Worms strains with the indicated reporters were exposed to each bacterial isolate. GFP expression was evaluated at 12 and 24 h of exposure with identical results. H69 cleavage and reporter expression is represented as in (D). **(G)** Fluorescent microscopy images of *hsp-4*::GFP worms (strain VL809) exposed to *E. coli* HB101 (nonpathogenic control) or *P. aeruginosa* wild isolate z23 for 24 h. Expression of *hsp-4*::*GFP* in the spermatheca is constitutively present and its indicated by white asterisks. All fluorescent micrographs show GFP expression level in the green color and expression of a pharynx labeling mCherry array marker in red (for AU133 strain). S3 Table contains source data for panels D and F. DIC, differential interference contrast; e. vector, empty vector; GFP, green fluorescent protein; H69, helix 69; wt, wild-type.

P38 MAP kinase pathways [15] were monitored using the *hsp-4*::GFP, *hps-6*::GFP, and *T28B8.5*::GFP reporter strains, respectively. Adult worms carrying these reporters were exposed to 5 representative *P. aeruginosa* isolates (3 of which elicit H69 cleavage and 2 of which do not, while all 5 selected to be of similar virulence), and GFP fluorescence was quantified after 24 h (Fig 4F and 4G, S3 Table and S6 Fig). The *hps-4*::GFP (UPR^ER) and *T28B8.5*::GFP (*pmk-1*/P38) reporters were induced in worms exposed to all 5 isolates. In contrast, *hsp-6*::GFP (UPR^mt) expression was absent. These results show that across these 5 representative *P. aeruginosa* isolates, the capacity to induce H69 cleavage is not correlated with activation of the worm's UPR^ER, UPR^mt, or *pmk-1*/P38 pathways. By contrast, the strong statistical association between rRNA H69 cleavage and *zip-2* pathway activation by *P. aeruginosa* wild isolates indicates that H69 cleavage of ribosomes is associated with a major translational repression mechanism in virulent strains of *P. aeruginosa*.

## Discussion

In this study, we sought to understand how virulent *P. aeruginosa* effects the host protein synthesis machinery, with particular focus on the host rRNAs. We found that upon interaction with PA14, *C. elegans* adult hermaphrodites experience the accumulation of ribosomes cleaved at their decoding center, specifically in the loop of 26S rRNA H69 (Fig 1A–1D). Cleavage of rRNAs at various sites of the large subunit rRNA have been reported in response to stress [22–24] and cell death conditions [25,41,42]. However, accumulation of ribosomes cleaved at the decoding center, particularly in the loop of H69, has not been previously described. Notably, we observed that H69 cleavage occurs independently of exotoxin A (Fig 2B and 2C), suggesting that H69 cleavage reflects part of a second bacterial virulence pathway to repress eukaryotic host translation. The supposition that cleavage of 26S rRNA at H69 should compromise protein synthesis is supported by previous findings that alteration of *E. coli* H69 impairs several steps of translation [32].

Analysis of the relationship between H69 cleavage and a host response to translation inhibition, namely activation of the *zip-2* pathway (i.e., *irg-1*::GFP expression), further supports the idea of H69 cleavage–induced protein synthesis inhibition. Genetic conditions that reduce or abrogate H69 cleavage also reduce or abrogate *zip-2* activation. Bacterial mutants of virulence regulators (e.g., *gacA*) abrogate both cleavage and activation of *zip-2* (Figs 2A and 4B). Similarly, the host *dyn-1* gene is required for both cleavage and activation of *zip-2* (Figs 2F and 4C). In addition, we find that among a set of *P. aeruginosa* wild strains, activation of the *C. elegans* *zip-2* pathway is strictly correlated with the induction of H69 cleavage (Fig 4D and 4E).

Although the *zip-2* pathway has been shown to respond to translation inhibition [11,20], *zip-2* pathway activation can be associated with mitochondrial dysfunction, which can elicit expression both the *zip-2* target *irg-1* and the UPR$^{mt}$ chaperone *hsp-6* [19,43,44]. However, our findings of an absence of *hsp-6* induction upon exposure to *P. aeruginosa* strains that elicit H69 cleavage (Fig 4F) disfavors the idea of mitochondrial dysfunction as main inducer of the *zip-2* pathway in this context.

The level of accumulation (relative to the total 26S pool) of H69-cleaved ribosomes can be appreciable. In the intestine, the organ in closer contact with the bacteria, about half of the ribosomal 26S RNA, could be cleaved by 24 h of bacterial exposure (Fig 3B). The fact that a sizable fraction of the infected worm's ribosomes become cleaved and hence likely translationally impaired suggests that H69 cleavage is a major contributor to virulence-associated translational repression by PA14. It remains to be determined to what extent other rRNA cleavage events, such as the less abundant H79 cleavage that we observed (Fig 1D), may contribute to PA14-induced translational repression.

Concomitantly with H69 cleavage, PA14-exposed worms undergo a drastic loss of rRNA (S1 Fig), yet our results show that H69 cleavage is not a mere consequence of pathogen-induced or starvation-induced reduction in organismal rRNA content (S1 Fig). These observations do not rule out that PA14-induced rRNA reduction and H69 cleavage may potentially be mechanistically linked. For example, they could both be the result of deregulated nuclease activity. Moreover, the reduction in worm ribosomal content upon PA14 infection, along with H69 cleavage, may contribute to translation inhibition in the host. Further studies could address the relative contribution of H69 cleavage and rRNA reduction for protein synthesis inhibition and *P. aeruginosa* virulence.

Among the bacteria that we tested here, which included a panel of wild isolates of *P. aeruginosa*, other *Pseudomonas* species, and several non-*Pseudomonas* pathogens, the ability to elicit H69 cleavage in *C. elegans* 26S rRNA was limited only to certain isolates of *P. aeruginosa* (S1A and S1C Fig). This finding suggests that the induction of H69 cleavage in host rRNA likely requires bacterial virulence factors relatively specific to *P. aeruginosa*. The existence of natural variation among *P. aeruginosa* isolates in their capacity to induce ribosome cleavage (S1C Fig) may be related to the fact that *P. aeruginosa* is a facultative pathogen of diverse host species. The capacity to induce cleavage may not be advantageous under all conditions, depending for example on the specific host and/or other factors.

Several lines of evidence support the idea that H69 rRNA cleavage is a component of *P. aeruginosa* virulence towards *C. elegans*. First, the accumulation of H69-cleaved rRNA tracks with the length of time adult worms are exposed to the bacteria, concomitantly with the progress of the bacterial infection (Fig 3A). Second, *P. aeruginosa* isolates that induce cleavage of the worm ribosomes are significantly more virulent than those who do not induce H69 cleavage (Fig 2B). Third, worms carrying mutations in the *pmk-1*, *zip-2*, and *fshr-1* genes, which impair 3 distinct pathways required for resisting PA14 infection, show both reduced survival on PA14 [15,17,19] and enhanced levels of H69 cleavage (Fig 2E). Finally, the capacity of *P. aeruginosa* strain PA14 to induce H69 cleavage in *C. elegans* requires certain genes (*rhlR*, *lasR*, *gacA*, *gacS*, and PA14_27700) that encode critical regulators of diverse virulence programs (Fig 2). Interestingly, our genetic tests of genes downstream of these virulence regulatory programs indicated that none of the factors tested (*exoU*; genes of the pyoverdin, rhamnolipid, phenazine, and pyochelin pathways; types I, II, III, V, and VI secretion systems) are required for H69 cleavage (S4 Fig). Therefore, we conclude that H69 cleavage depends on unidentified virulence factors downstream of the QS and 2 component regulators.

H69 rRNA cleavage in *C. elegans* adults exposed to *P. aeruginosa* appears to not simply reflect degradative processes associated with cell or organismal death. Firstly, H69 ribosome

cleavage starts to manifest in worms after less than 8 h of exposure to PA14, when no cellular or organismal indications of death are evident (Fig 3A). Moreover, H69 cleavage is not dependent on the apoptotic program, as the cleavage occurs unimpeded in the *ced-3(n717)* mutants (S5A Fig). Similarly, cleavage is not observed upon treatment of worms with cry6A, a necrosis inducer (S1A Fig). Another observation indicating that H69 cleavage is not a consequence of cell or organismal death is our finding that *C. elegans* larvae growing on PA14 exhibit H69 cleavage despite the full resistance of larvae to PA14 infection (S5B Fig).

Our observation that H69 cleavage is abrogated in worms with dynamin knockdown, which encodes a protein essential for endocytosis, suggests that cleavage may be elicited by a nuclease that is secreted by *P. aeruginosa* and actively internalized into *C. elegans* intestinal cells. On the other hand, we cannot currently rule out the possibility that rRNA cleavage occurs in response to uptake of a bacterially encoded trigger that disrupts the RNA targeting specificity of a host-encoded nuclease. Our observation that H69-cleaved 26S rRNA fragments are evident in the 80S and heavier fractions of sucrose gradient fractionated lysates of infected worms suggests that H69 cleavage may occur within intact 60S subunits and/or perhaps even within assembled 80S::mRNA complexes. The supposition that intact ribosomes could be the substrate for cleavage in vivo is further supported by our finding that an S100 extract of PA14-infected worms contain H69-cleaving activity against purified *C. elegans* ribosomes. Further studies are required to identify the molecular identity of the nuclease responsible for H69 cleavage in *C. elegans* and to characterize its substrate specificity, mechanisms of action, and biological functions in various contexts.

## Materials and methods

### *C. elegans* strains

The *C. elegans* N2 strain was used as a wild-type strain. All nematode strains were maintained using standard methods on "Nematode Growth Media" (NGM) plates [45] and fed with *E. coli* HB101. The *C. elegans* strains used in the present study are listed in the S1 Table.

### Bacterial strains

All bacterial strains (*P. aeruginosa*, *E. coli*, *E. faecalis*, etc.) were routinely grown on Lysogeny Broth (LB) media at 37°C without antibiotics, unless otherwise noted. A list of the bacterial isolates employed in the present study is found in S2 Table. Transposon insertion mutants of *P. aeruginosa* were obtained from the PA14NR library [46].

### *C. elegans*–*P. aeruginosa* interaction assays

Exposure of *C. elegans* to *P. aeruginosa* was performed using SK conditions [12]. Briefly, an aliquot of an overnight liquid LB culture of *P. aeruginosa* was plated on SK agar plates. The bacterial lawn was spread to cover the entire surface of the agar and to prevent worms from easily escaping the bacterial lawn. The plates were incubated at 37°C for 24 h and then at 25°C for 24 h, to allow growth of the lawn and the induction of pathogenic activity [12]. A synchronous population of *C. elegans* worms was prepared by standard hypochlorite treatment, followed by culture of larvae from L1 stage to required developmental stage on NGM agar plates seeded with *E. coli* HB101. Worms were grown to experimentally chosen stages and then transferred to the SK plates to initiate their exposure to *P. aeruginosa* lawns. For most assays, synchronized young adults (YAs) were used, as exception, for the assay in S3B Fig, synchronized L1 larvae of stain VT1367 were employed. VT1367 carries a transgene (*col-19*::*gfp)* that indicates the larva to adult transition in an otherwise wild-type background.

## Exposure of *C. elegans* to abiotic stresses and bacteria

Exposure of worms to other tested *Pseudomonas* species (S1A Fig) was carried out on conditions similar to the *P. aeruginosa* assays (see above) with the following modification: After bacterial seeding, the SK plates were incubated at 25˚C for 48 h, as most tested *Pseudomonas* species do not grow at 37˚C (of note, on this modified temperature regime, *P. aeruginosa* PA14 also induces H69 cleavage). YA worms were exposed to the bacteria for 24 h then collected, and their RNA was extracted and analyzed as done for the *P. aeruginosa* assays. Exposure of *C. elegans* to *S. aureus*, *S. enterica*, and *E. faecalis* proceed as documented by Powell and Ausubel [47].

Worm were exposed to a set of drugs (S1A Fig) by direct addition of the drugs to lawns of *E. coli* HB101 to a determined final concentration. Following drug supplementation of the lawns, YA worms were added and exposed for 24 h, then processed as done for the *P. aeruginosa* assays. Specific conditions for the used drugs and other abiotic stresses are as follows: DTT (2.5 mM), Tunycamicin (5 μg/mL), cycloheximide (500 μg/mL), harringtonine (63 μg/mL), blasticidin S (250 μg/mL), cadmium chloride (6 mM), and sodium arsenite (1 and 5 mM).

## Analysis of worm RNA profiles

After the worms were exposed to *P. aeruginosa* for the experimentally designated times (ranging from 0 to 24 h), they were collected in M9 buffer and decanted 3 times in 15-mL tubes to allow digestion of bacteria inside the worms and to separate adults from any larval progeny (no 5-Fluoro-2′-deoxyuridine was utilized in the *P. aeruginosa* assays). Following washing, the worm pellets were treated with guanidinium thiocyanate and RNA isolated using phenol/chloroform and ethanol precipitation. The obtained RNA was quantified and subjected to capillary electrophoresis using a 5300 Fragment Analyzer system (Agilent, California, United States of America) using a "standard sensitivity" RNA assay. The obtained electropherograms were analyzed using the ProSize 2.0 software (Agilent) to quantify the relative concentration of distinct rRNA species.

For the analysis of the total RNA content per worm (S1B Fig), sets of 20 adult worms were exposed to the designed treatment for 24 h. Following exposure, the worms were washed with M9 buffer, and individual worms were manually transferred into 0.25-mL guanidinium thiocyanate containing a fixed amount of a 500-nt spike-in RNA (in vitro translated). RNA was isolated by phenol–chloroform extraction followed by ethanol precipitation. RNA was profiled with capillary electrophoresis as described above.

## RNA excision and cloning

For the analysis of the worm rRNA fragments induced by *P. aeruginosa* (Fig 1A and 1B), total RNA from worms exposed to PA14 for 24 h, was separated by electrophoresis in a 4.5% polyacrylamide gel electrophoresis (PAGE)–Urea denaturing gel (National Diagnostics, Georgia, USA), and the 2 PA14-induced bands were excised and smashed in 100 μL of buffer (198-mM KCl, 2-mM Tris pH 7.9, and 0.2-mM EDTA) and incubated for 90 min at 60˚ C to allow the RNA to diffuse out, followed by extraction with phenol–chloroform and ethanol precipitation. To determine 3′ termini, RNA was treated with T4 polynucleotide kinase + ATP, and a 3′ adenylated linker RNA was ligated using T4 RNA ligase 2. To determine 5′ termini, a non-phosphorylated linker RNA was ligated with T4 RNA ligase 1. For all samples, ligation was followed by random hexamer primed cDNA synthesis using Superscript III reverse transcriptase (Thermo Fisher Scientific, Massachusetts, USA). PCR amplification was carried out with

primers complementary to the 3′ linker and multiple positions along the 26S rRNA. The amplicons obtained were then Sanger sequenced directly or after TOPO cloning.

### Degradome sequencing

Total RNA from worms exposed to *P. aeruginosa* PA14 for 24 h was isolated using guanidinium thiocyanate with phenol–chloroform and ethanol precipitation. A non-phosphorylated linker RNA was ligated to the RNA 5′ termini using T4 RNA ligase 1. Ligated RNA was reverse transcribed using a DNA adaptor primer (carrying an 8-nt randomer) and Superscript III reverse transcriptase (Thermo Fisher Scientific). cDNA was treated with RNAse A and H, then second strand synthesis was conducted with a primer complementary to the 5′ linker sequence and Q5 high fidelity DNA polymerase (NEB, Massachusetts, USA). The 300- to 500-bp fragment range of the obtained dsDNA was selected on an agarose gel and purified. The obtained DNA was PCR amplified for 12 rounds and then subjected to Illumina sequencing (Amplicon-EZ service, Genewiz, New Jersey, USA).

### Analysis of degradome sequencing data

Sequencing reads from the degradome libraries were filtered for the presence of a complete 5′ adaptor sequence (5′-TCTACAGTCCGACGATCTGAC) using the tool "cutadapt." The read's 3′ adaptor was similarly trimmed. The filtered reads were then mapped to the *C. elegans* rRNAs using the "BWA" software [48].

Reads with identical 5′ terminus were counted altogether in order to determine the corresponding 5′ terminus frequency. The number of total reads mapped to each rRNA species (18S, 26S, 5.8S, or 5S) was used to normalize the 5′ termini frequency relative to 1 million reads. For the 2 sequencing libraries, a similar number of reads were mapped to the 26S rRNA (49,491 for the library of PA14-exposed worms and 33,777 for the library of HB101-exposed worms). For the 26S-mapped reads, an outlier "background" line was established; the line separates the 0.7% of reads with the highest counts. The identified 26S 5′ termini are reported on S4 Table.

### Polysome profiling

YA wild-type worms were exposed to *P. aeruginosa* PA14 or *E. coli* HB101 for 24 h, then washed with M9 buffer and pelleted. The worms were lysed using a Dounce grinder and lysis buffer (10-mM Tris-HCl pH 7.5, 5-mM MgCl$_2$, 100-mM KCl, 1% Triton X-100, 2-mM DTT, 100-μg/ml cycloheximide, EDTA-free protease inhibitor tablet (Roche, Indiana, USA), and SUPERase-In RNAse inhibitor (Thermo Fisher Scientific)). The lysates were then cleared by centrifugation at 2,000 revolutions per minute (rpm) for 10 min. The cleared lysates were loaded on top of a 10% to 50% (weight/volume) sucrose gradient and centrifuged at 35,000 rpm in a SW-40Ti rotor for 3.5 h. The gradient was then fractionated while measuring absorbance (A260) and 16 fractions collected. RNA from the gradient fractions was extracted using guanidinium thiocyanate with phenol–chloroform and ethanol precipitation. RNA was separated by electrophoresis in a 4.5% PAGE–Urea denaturing gel (National Diagnostics) and stained with SYBR-Gold (Thermo Fisher Scientific).

### RNA-mediated interference (RNAi)

RNAi by feeding was performed using *E. coli* strains expressing a plasmid vector targeting *dyn-1* or empty vector. The bacteria were grown on LB ampicillin media and plated on NGM

plates with 1-mM IPTG. The seeded plates were induced at 25˚ C for 24 h. Synchronized YA worms were then put on the plates to start the experiments.

## Bacterial ToxA expression

*E. coli* strains expressing a plasmid vector with the *toxA* gene or an empty vector were used following previously reported conditions [11]. The bacterial strains were grown on LB ampicillin, induced with IPTG for 1 h, and then plated on NGM plates with 1-mM IPTG. The seeded plates were induced at 25˚ C for 24 h. Synchronized YA worms were then put on the plates to start the experiments.

## Ribosome immunoprecipitation (IP)

Strains expressing *rpl-1*::GFP with tissue specific promoters (JAC127 and JAC382) [31] were used to determine H69 cleavage levels in the worm intestine and muscle (Fig 2B). The transgenic worms were exposed to PA14 under standard conditions (as described above), collected, and pelleted. Worm lysates were prepared using a Dounce tissue grinder and homogenization buffer (50-mM Tris pH 7.5, 100-mM KCl, 12-mM MgCl$_2$, 0.1% Triton X-100, PMSF 1-mM, EDTA-free protease inhibitor tablet (Roche), and SUPERase-In RNAse inhibitor (Thermo Fisher Scientific)). The lysates were cleared by centrifugation at 9,000 rpm for 20 min. Protein G magnetic beads (dynabeads, Thermo Fisher Scientific) were coupled to an anti-GFP antibody (TP401, Torrey Pines) for 1 h. GFP from the lysate was immunoprecipitated with the coupled beads for 30 min, followed by 4 washing steps with cold high salt buffer (50-mM Tris pH 7.5, 300-mM KCl, 2-mM MgCl$_2$, and 0.1% Triton X-100). All steps were carried out at 4˚ C. Guanidinium thiocyanate was added to the immunoprecipitated samples and RNA isolated using phenol–chloroform and ethanol precipitation.

## In vitro ribosome cleavage assay

**Preparation of S100 extracts.** Synchronized adult *C. elegans* were exposed 24 h to PA14 using SK conditions. The worms were collected, then washed with M9 buffer and pelleted. The *C. elegans* were lysed using a Dounce grinder and complete lysis buffer (30-mM HEPES pH 7.4, 2-mM MgCl$_2$, 50-mM KCl, 0.1% Triton X-100, 5% glycerol, 2.5-mM DTT, 1-mM PMSF, EDTA-free protease inhibitor tablet (Roche), and SUPERase-In RNAse inhibitor (Thermo Fisher Scientific)). The lysates were then cleared by centrifugation at 3,300 rpm for 5 min and 12,700 rpm for 20 min. The cleared lysates were ultracentrifugated in an Optima MAX-TL centrifuge using a TLA-100 rotor for 1 h at 55,000 rpm. The obtained supernatant were maintained (S100 extract). A control S100 extract was prepared from synchronized worms maintained on standard nonpathogenic conditions (i.e., on *E. coli* HB101) following the above procedure.

**In vitro assay.** *C. elegans* expressing GFP-tagged ribosomes (JAC382 strain) were maintained on standard nonpathogenic conditions. The worms were used to immunoprecipitate ribosomes following the procedure described in the "Ribosome immunoprecipitation (IP)" section. An assay reaction was constituted by S100 extract and purified ribosomes in reaction buffer (50-mM potassium acetate, 20-mM Tris-acetate, 10-mM magnesium acetate, 100-μg/ml BSA, pH 7.9, and murine RNAse inhibitor 1 U/microL). The assay were incubated at 26˚C for 60 min, then quenched on ice. The assay samples were treated with guanidinium thiocyanate and RNA isolated using phenol/chloroform and ethanol precipitation. The RNA was analyzed by capillary electrophoresis as described in the "Analysis of worm RNA profiles" section.

## Microscopy

Microscopic analysis of transgene fluorescence (GFP or mCherry) was carried out using a Zeiss Imager Z1 microscope (Zeiss, Germany) equipped with an Axiocam 503 camera and Zen Blue software (Zeiss). The worm strains of *irg-1*::*GFP*, *hsp-4*::*GFP*, *hsp-6*::*GFP*, and *T24B8,5*::*GFP* were maintained on *E. coli* at 20°C. These transgenic worms were synchronized by hypochlorite treatment and exposed as YAs to strains of *P. aeruginosa* under standard assay conditions. Worm images were taken with a 10× objective and constant exposure settings: 200 ms for the GFP and dsRed fluorescent channels. Images were exported and further processed with ImageJ [49,50]. A summary of the reporter results, strain virulence, and capacity to induce H69 cleavage is found in S3 Table.

## Supporting information

**S1 Fig. Specificity of H69 cleavage induction by different stresses and bacteria. (A)** Matrix summary of H69 cleavage induction by 28 different stresses and bacteria. Worms were exposed to the indicated condition for 24 h at 25°C, and worm total RNA was extracted and analyzed by capillary electrophoresis. Only PA14 induces cleavage at H69 (indicated by green box). **(B)** Matched measurement of H69 cleavage (left graph) and total RNA content (right graph) for wild-type worms exposed 24 h to either *Pseudomonas aeruginosa* PA14, *Escherichia coli* HB101, *Enterococcus faecalis* (*E.f.*), and starvation (starved). "∗∗" indicates *p*-value < 0.01 for Welch *t* test comparison to *P. ae.* (left graph) or *E. coli* (right graph). S1 Data contains source data for panel B. **(C)** Collection of *P. aeruginosa* isolates tested for their capacity to induce H69 cleavage on worms (H69 cleavage–inducing capacity indicated by red color). The phylogenetic tree of the isolates was obtained from [27].
(TIF)

**S2 Fig. In vitro ribosome cleavage. (A)** RNA profile of the S100 extract from PA14-exposed worms, used for the in vitro assays (top panel); RNA profile of ribosomes after incubation with S100 extract from control (middle panel) or PA14-exposed worms (bottom panel). Arrowheads indicate bands at approximately 1,100 and approximately 2,300 nt. "M" indicates a 15-nt marker used in the electrophoretic separation system. **(B)** Termini of the 2 bands in (A) are indicated by arrowheads. The termini map to the H69 of 26S rRNA. The number of independent sequences mapped to the cut sites is shown in parenthesis.
(TIF)

**S3 Fig. 5′ rRNA termini of worms exposed to PA14.** Frequency of 5′ termini along the worm 18S rRNA **(A)**, 5S rRNA **(B)**, 5.8S rRNA **(C)**, and 26S rRNA **(D)** as determined by degradome sequencing. The H69 cleavage in the 26S rRNA is indicated by the black arrowhead.
(TIF)

**S4 Fig. Induction of H69 cleavage in *P. aeruginosa* virulence-related mutants.** Matrix summary of H69 cleavage induction by a set of *P. aeruginosa* mutants in virulence genes (top row) and secretion systems (lower row). PA14 wt and *gacA* are included as controls. Worms were exposed to the indicated mutant bacteria for 24 h at 25°C, and worm total RNA was extracted and analyzed by capillary electrophoresis. Induction of H69 cleavage is indicated by the box color (green: cleavage level similar to PA14 wild type; white: absent cleavage).
(TIF)

**S5 Fig. H69 cleavage in conditions of abrogated cell death. (A)** H69 cleavage levels for a *ced-3* loss of function *(lf)* mutant, which is defective in apoptosis. The cleavage levels are not significantly different (n.s.) than those of wild-type (wt) worms (Welch *t* test comparison, *p*-

value > 0.05). **(B)** Graph of H69 cleavage levels in relationship to larval worm exposure time to PA14 (in hours). L1 larvae are experimentally set to start development on PA14 ($t = 0$) and transition from larvae to adult (L/A) at the time indicated by the dashed blue line (average time in hours). The average of tree measurements per timepoint and standard deviation (error bars) is plotted. S1 Data contains source data for all panels.
(TIF)

**S6 Fig. Expression of host response pathways upon exposure to *P. aeruginosa*.** Fluorescent microscopy images of *hsp-6*::*GFP* worms exposed to PA14 or *Escherichia coli* HB101. Young adult worms of the indicated reporter were exposed to the respective bacteria for 24 h. GFP expression level is indicated by the green color. Nomarski images (DIC) corresponding to the fluorescent pictures are shown.
(TIF)

**S1 Data. Numerical data and statistics for selected plots.** Separate sheets are named after the corresponding figure panel. Each sheet contains the numerical values and statistics used for the corresponding plots.
(XLSX)

**S1 Table. *C. elegans* strains.**
(XLSX)

**S2 Table. Bacterial strains.**
(XLSX)

**S3 Table. Features of the *P. aeruginosa* collection.**
(XLSX)

**S4 Table. Identified 26S 5′ termini.**
(XLSX)

## Acknowledgments

We would like to acknowledge Micah Belew and members of the Ambros and Mello laboratories for feedback about this research project. Some of the investigated bacterial strains were obtained from International Health Management. Regarding *C. elegans*, some strains were provided by the *Caenorhabditis* Genetics Center (CGC).

## Author Contributions

**Conceptualization:** Alejandro Vasquez-Rifo, Victor Ambros.

**Data curation:** Alejandro Vasquez-Rifo.

**Formal analysis:** Alejandro Vasquez-Rifo.

**Funding acquisition:** Victor Ambros.

**Investigation:** Alejandro Vasquez-Rifo, Emiliano P. Ricci, Victor Ambros.

**Methodology:** Alejandro Vasquez-Rifo, Emiliano P. Ricci, Victor Ambros.

**Resources:** Alejandro Vasquez-Rifo.

**Software:** Alejandro Vasquez-Rifo.

**Supervision:** Victor Ambros.

**Validation:** Alejandro Vasquez-Rifo, Victor Ambros.

**Visualization:** Alejandro Vasquez-Rifo.

**Writing – original draft:** Alejandro Vasquez-Rifo, Emiliano P. Ricci, Victor Ambros.

**Writing – review & editing:** Alejandro Vasquez-Rifo, Emiliano P. Ricci, Victor Ambros.

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
