## [Editor Report · Decision Letter 0]

20 Jul 2020

Dear Dr Vasquez-Rifo, 

Thank you for submitting your manuscript entitled "Pseudomonas aeruginosa cleaves the decoding center of Caenorhabditis elegans ribosomes" for consideration as a Research Article by PLOS Biology. Thank you also for your patience as we completed our editorial process, and please accept my apologies for the delay in providing you with our decision - the last two weeks have been extremely busy.

Your manuscript has now been evaluated by the PLOS Biology editorial staff as well as by an academic editor with relevant expertise and I am writing to let you know that we would like to send your submission out for external peer review.

Please re-submit your manuscript within two working days, i.e. by Jul 22 2020 11:59PM.

Kind regards,

Ines

--

Ines Alvarez-Garcia, PhD

Senior Editor

PLOS Biology

---

## [Decision Letter · Decision Letter 1]

17 Sep 2020

Dear Dr Vasquez-Rifo,

Thank you very much for submitting your manuscript "Pseudomonas aeruginosa cleaves the decoding center of Caenorhabditis elegans ribosomes" for consideration as a Research Article at PLOS Biology. Thank you also for your patience as we completed our editorial process, and please accept my apologies for the delay in providing you with our decision. Your manuscript has been evaluated by the PLOS Biology editors, an Academic Editor with relevant expertise, and by four independent reviewers.

You will see that the reviewers find your manuscript interesting and important for the field, however they also think the results are a bit preliminary and the mechanistic aspect needs to be strengthened by identifying the nuclease involved in the process. After discussing the reviews with the academic editor and the rest of the team, we have decided to offer you the choice of either revising the manuscript to add the missing mechanistic insights that would complete the story as a Research Article, or resubmit it as a Discovery Report addressing the minor points raised by the reviewers. This last option would allow you to continue your research to identify the nuclease and submit it as an ‘Update’ article in the near future.

In light of the reviews (attached below), we will not be able to accept the current version of the manuscript, but we would welcome re-submission of a revised version that takes into account the reviewers' comments accordingly with the two options we offer. We cannot make any decision about publication until we have seen the revised manuscript and your response to the reviewers' comments. Your revised manuscript is also likely to be sent for further evaluation by the reviewers.

We expect to receive your revised manuscript within 3 months. 

**IMPORTANT - SUBMITTING YOUR REVISION**

*Re-submission Checklist*

*Published Peer Review*

*PLOS Data Policy*

*Blot and Gel Data Policy*

Best wishes,

Ines

--

Ines Alvarez-Garcia, PhD,

Senior Editor,

ialvarez-garcia@plos.org,

PLOS Biology

Reviewers’ comments

Rev. 1:

In "Pseudomonas aeruginosa cleaves the decoding center of Caenorhabditis elegans ribosomes," the authors show that the pathogen P. aeruginosa inhibits translation in the host (C. elegans) in part by a very specific cleavage (H69) of the 26S ribosomal RNA. Previous work had shown inhibition of translation by P. aeruginosa in a general manner via the action of Exotoxin A (ToxA). However, this new finding is novel and occurs independently of ToxA. While the exact bacterial factors involved as well as the mechanism are not identified, the authors show that the activity is dependent on important virulence/quorum sensing regulators such as GacA. They also shown that the process is dependent on host dynamin, possibly because the bacterial factor(s) involved are brought in through the endocytosis pathway. Finally, the authors demonstrate that three major host defense pathways, mediated by PMK-1, FSHR-1 and ZIP-2 antagonize this cleavage. The technical aspects of the manuscript are very strong, and the story is clear and compelling. Identification of this new way in which P. aeruginosa manipulates host physiology is an important finding. However, the manuscript would be strengthened by further insight into the identity of the nuclease and whether this happens in hosts other than C. elegans.

Major Comments:

1) Identification of the nuclease and/or more of the mechanism would really make this story.

2) Is there any evidence that this fascinating mechanism of translational inhibition occurs in mammalian systems? For example, have you looked for this cleavage event after infecting lung epithelium cells with PA14?

3) Is the dynamin mutant/RNAi less sensitive to killing by P. aeuruginosa, as would be predicted?

Minor Comments:

1) The statement in line 47-48 is a bit misleading. A mutation in the toxA gene does appreciably alter virulence towards certain C. elegans mutant backgrounds (such as zip-2). Perhaps one can say specifically that it does not alter virulence for "wild type" or "N2."

2) Line 161: Should be Figure S2C.

Rev. 2:

For this revised MS, I have not seen the reviewers' comments on the previous version, nor the authors' rebuttal.

This is a clear and logical examination of the potential for PA14 to provoke specific cleavage of 26S rRNA in C. elegans. It represents a real step forward in our understanding of a virulence strategy and how a surveillance-type immune response is triggered. Even in the absence of mechanistic details (both on the pathogen and host sides) it will be of relatively broad interest.

I only have very minor comments.

"Several lines of evidence support the idea that H69 rRNA cleavage is a component of P. aeruginosa virulence towards C. elegans…..Third, worms carrying mutations in the pmk-1, zip-2 and fshr-1 genes, which impair three distinct pathways required for resisting PA14 infection, show both reduced survival on PA14 [15,17,19] and enhanced levels of H69 cleavage (Fig 3A).

As shown by Miller et al. (10.1371/journal.pone.0137403; not cited), for fshr-1, that reduced survival on PA14 is correlated with an increased bacterial load in the gut. The same is true for pmk-1 and zip-2. The enhanced levels of H69 cleavage here could thus very simply be secondary consequence and this does not to my mind represent a line of evidence.

I take issue with the first sentence that starts, "Pathogens such as P. aeruginosa adventitiously modify animal host physiology". It is unlikely that these modifications are adventitious. They are more likely to be the result of a long process of selection.

including nematodes such as C. elegans and vertebrates

including vertebrates and nematodes such as C. elegans

Rev. 3:

Vasquez-Rifo et al. characterize a mechanism by which pathogenic Pseudomonas aeruginosa inhibits protein synthesis of the host C. elegans. The authors demonstrate using capillary electrophoresis that exposure to P. aeruginosa results in the accumulation of ribosomes that have undergone specific cleavage at helix 69 of the 26S ribosomal RNA, a component of the ribosome decoding center. H69 cleavage is shown to activate ZIP-2-dependent signaling. Prior studies have shown that translational inhibition activates ZIP-2-dependent signaling, and the authors' results confirm this to be the case during H69 cleavage by P. aeruginosa. The cleavage is also independent of the previously characterized P. aeruginosa toxin ToxA.

The quantitative capillary electrophoresis analysis impressively demonstrates specific cleavage of the 26S rRNA, and experiments demonstrate the dependence of this cleavage on the virulence of the Pseudomonas aeruginosa, and the enhancement of the cleavage in genetic backgrounds in which pathogen resistance is impaired. Cleavage is specific to P. aeruginosa and is not observed during infection by other pathogenic bacteria. Notably, the authors also demonstrate a dramatic reduction in total RNA content during P. aeruginosa infection.

The bacterial or host enzyme catalyzing cleavage remains undefined. Interestingly, the cleavage event requires activity of DYN-1, suggestive of a dependence of cleavage on mechanisms of endocytic internalization. The results in the manuscript are of considerable interest, as one could envision a similar process occurring in cells of other host organisms under attack by P. aeruginosa, and bacterial virulence effector mechanisms continue to yield basic insights into both pathogen and host biology. The story would be greatly strengthened through identification of the bacterial factors involved. Many ribotoxic effectors produced by bacteria have been characterized, and one might anticipate either through sequence homology searching or direct analysis of the P. aeruginosa PA14 mutant library, further mechanistic insight might be derived. This is a lot to ask, but at the same time, simply the characterization of a cleavage event seems a bit preliminary, like the opening act of a more complete story.

Rev. 4:

In this manuscript, "Pseudomonas aeruginosa cleaves the decoding center of Caenorhabditis elegans ribosomes", the authors describe an extraordinary intriguing and exciting mechanism by which P. aeruginosa inhibits protein synthesis in its host C. elegans by cleaving the 26S rRNA specifically at helix 69. This cleavage of 26S rRNA at H69 results in the activation of the host defense response, through what appear to be novel mechanism(s). Although these latter data are more correlative, the authors convincingly show that H69 cleavage and host immune activation does not occur in a straightforward way through known mechanisms. Finally, the authors convincingly argue that the 26S rRNA cleavage is a mechanism by which the pathogen abrogates the host defense and antibacterial response. The manuscript describes novel, original and exciting biology. The experiments are well thought through and the results are convincing. The manuscript is very well written, and I personally appreciated the attentiveness with which the authors discussed their results.

I have only one minor concern regarding experiments. The authors conduct experiments that suggest that the 26S rRNA cleavage occurs at intact ribosomes. The authors conclude this from experiments depicted in S1A Fig showing that an rRNA-free S100 extract prepared from P. aeruginosa infected animals can cleave ribosome-associated rRNA from unexposed animals. These experiments would be more convincing if the authors included data showing the profile of the rRNA-free S100 prepared from P. aeruginosa infected animals prior to incubation with intact ribosomes.

Minor points

1) Figure S1 is discussed in the manuscript after Figure S2.

2) Line 174: missing an 'of'.

3) Line 358: should be S100.

---

## [Editor Report · Decision Letter 2]

6 Oct 2020

Dear Dr Vasquez-Rifo,

Thank you for submitting your revised Discovery Report entitled "Pseudomonas aeruginosa cleaves the decoding center of Caenorhabditis elegans ribosomes" for publication in PLOS Biology. I have now obtained advice from the Academic Editor and discussed your responses to the reviewers.

We're delighted to let you know that we're now editorially satisfied with your manuscript. However before we can formally accept your paper and consider it "in press", we also need to ensure that your article conforms to our guidelines. A member of our team will be in touch shortly with a set of requests. As we can't proceed until these requirements are met, your swift response will help prevent delays to publication. Please also make sure to address the data and other policy-related requests noted at the end of this email.

- a cover letter that should detail your responses to any editorial requests, if applicable

*Copyediting*

*Published Peer Review History*

*Early Version*

Sincerely,

Ines

--

Ines Alvarez-Garcia, PhD,

Senior Editor,

ialvarez-garcia@plos.org,

PLOS Biology

Fig. 1C; Fig. 2A, B, C, D, E F; Fig. 3A, B; Fig. 4D, F; Fig. S1B and Fig. S5A, B

For manuscripts submitted on or after 1st July 2019, we require the original, uncropped and minimally adjusted images supporting all blot and gel results reported in an article's figures or Supporting Information files. We will require these files before a manuscript can be accepted so please prepare and upload them now. Please carefully read our guidelines for how to prepare and upload this data: https://journals.plos.org/plosbiology/s/figures#loc-blot-and-gel-reporting-requirements.

---

## [Editor Report · Decision Letter 3]

22 Oct 2020

Dear Dr Vasquez-Rifo,

On behalf of my colleagues and the Academic Editor, Matthew K Waldor, I am pleased to inform you that we will be delighted to publish your Discovery Report in PLOS Biology. 

PRODUCTION PROCESS

Before publication you will see the copyedited word document (within 5 business days) and a PDF proof shortly after that. The copyeditor will be in touch shortly before sending you the copyedited Word document. We will make some revisions at copyediting stage to conform to our general style, and for clarification. When you receive this version you should check and revise it very carefully, including figures, tables, references, and supporting information, because corrections at the next stage (proofs) will be strictly limited to (1) errors in author names or affiliations, (2) errors of scientific fact that would cause misunderstandings to readers, and (3) printer's (introduced) errors. Please return the copyedited file within 2 business days in order to ensure timely delivery of the PDF proof. 

If you are likely to be away when either this document or the proof is sent, please ensure we have contact information of a second person, as we will need you to respond quickly at each point. Given the disruptions resulting from the ongoing COVID-19 pandemic, there may be delays in the production process. We apologise in advance for any inconvenience caused and will do our best to minimize impact as far as possible.

EARLY VERSION

PRESS 

Kind regards,

Erin O'Loughlin

Publishing Editor, 

PLOS Biology

on behalf of

Ines Alvarez-Garcia,

Senior Editor

PLOS Biology